# Improving Accessibility to Patients with Interstitial Lung Disease (ILD): Barriers to Early Diagnosis and Timely Treatment in Latin America

**DOI:** 10.3390/ijerph21050647

**Published:** 2024-05-19

**Authors:** Ricardo G. Figueiredo, Nathalia Filgueiras Vilaça Duarte, Daniela Carla Barbosa Campos, Manuel de Jesus Diaz Verduzco, Ángel Alemán Márquez, Gabriela Tannus Branco de Araujo, Adalberto Sperb Rubin

**Affiliations:** 1Programa de Pós-Graduação em Saúde Coletiva, Universidade Estadual de Feira de Santana (UEFS), Feira de Santana 44036-900, Brazil; 2Boehringer Ingelheim do Brasil Química e Farmacêutica, São Paulo 04794-000, Brazil; nathalia.duarte@boehringer-ingelheim.com (N.F.V.D.); daniela.campos@boehringer-ingelheim.com (D.C.B.C.); 3Hospital Regional “Dr. Manuel Cardenas de la Vega”, Instituto de Seguridad y Servicios Sociales de los Trabajadores del Estado (ISSSTE), Culiacán Rosales 80230, Mexico; manydv@yahoo.com.mx; 4Hospital Naval de Especialidades de Veracruz, Hospital Español Veracruz, Universidad del Valle de México (UVM), UNAM Campus Veracruz, Veracruz 91700, Mexico; doctor.aleman.marquez@gmail.com; 5Axiabio Life Sciences International, São Paulo 04038-032, Brazil; gabriela.tannus@axia.bio.br; 6Santa Casa Hospital Porto Alegre/UFCSPA, Porto Alegre 90035-074, Brazil; arubin@terra.com.br

**Keywords:** interstitial lung diseases, patient journey, Latin America

## Abstract

Delayed initiation of effective antifibrotic therapy in patients with interstitial lung diseases (ILD) may influence the progression and outcome of the disease. This study analyzes the differences in the journey of patients with ILD in the Brazilian and Mexican health systems. An evaluative study was conducted in reference centers for interstitial lung diseases in Brazil and Mexico with a panel of four specialists. The patient’s journey in both countries begins when the patient seeks medical care after observing a chronic respiratory symptom. In both countries, due to diagnostic complexity, these patients arrive at ILD referral centers at an advanced stage of the disease. Once diagnosis is established, the treatment onset differs between Mexico and Brazil. In Brazil, access to antifibrotic drugs through the public health system has been a significant challenge, and their cost makes them unaffordable for most people. This situation forces medical specialists to provide only supportive care to patients until these drugs can be accessed. In Mexico, antifibrotics have been available in health sectors since 2018. Brazil and Mexico have several similarities regarding the initial journey of the patient due to diagnosis difficulties. Still, the outcome tends to be different due to a difference in access to treatment with antifibrotics. For this reason, advancing health policies that ensure proper treatment for patients with ILD is crucial for the sustainability and reliability of the health system.

## 1. Introduction

Interstitial lung disease (ILD) is a term that encompasses a range of diseases characterized by inflammation and fibrosis of the lung parenchyma. Despite their shared characteristics, these diseases vary significantly in their clinical presentations, underlying pathologies, prognosis, and responses to treatment [1]. The etiologies of ILDs include systemic diseases, environmental exposures, and idiopathic origins [2]. Understanding which pulmonary structures (small airways, alveoli, and vasculature) are primarily affected, in addition to the interstitial involvement pattern, can aid in diagnosing and classifying specific ILDs, guiding appropriate treatment strategies.

Essentially, interstitial pneumonias are characterized by varying degrees of inflammation and fibrosis, determined by individual characteristics and the pathophysiology of the disease. Notably, a strong correlation exists between high-resolution CT (HRCT) patterns and pathological features derived from lung biopsies. Idiopathic pulmonary fibrosis (IPF) is an epithelial–fibroblastic disorder where unknown stimuli disrupt alveolar epithelial cells’ homeostasis, leading to abnormal epithelial cell repair and fibroblast activation [3,4]. This results in excessive extracellular matrix accumulation, culminating in irreversible lung parenchymal destruction. Delayed fibroblasts and myofibroblasts apoptosis during alveolar epithelial repair play a significant role in IPF [5].

The ILD category comprises idiopathic interstitial pneumonia, autoimmune ILD, drug-associated ILD, complications from radiation therapy, occupational exposures, infection-related ILD, hypersensitivity pneumonitis (HP) and unclassifiable ILD [1,6]. Epidemiological studies showed that IPF is less prevalent than HP and connective tissue disease-related ILDs (CTD-ILD) [7,8,9]. IPF predominantly affects older men and has a median survival rate of 2.9 years without specific treatment, whereas CTD-ILD tends to be more prevalent among younger patients [10,11,12]. Interstitial involvement in collagen diseases is associated with a worse prognosis and is indeed the most frequent cause of death in Systemic Sclerosis (SSc) [13,14].

Distinguishing the various presentations of ILDs is a key factor in defining appropriate treatment and prognosis. Understanding the distinct clinical presentations is crucial for diagnosis, often requiring a multidisciplinary approach for comprehensive clinical, radiological, and pathological correlation [2,15,16]. Early diagnosis is based on the decline in lung function and HRCT abnormalities in the appropriate clinical setting [15]. In the age of antifibrotic therapies, it is indisputable that imaging tests play a fundamental role in early diagnosis and prognosis. The diagnostic accuracy of HRCT is high enough to detect even subclinical changes that occur in early-stage ILD [16]. ILD histologic patterns include (i) usual interstitial pneumonia (UIP); (ii) nonspecific interstitial pneumonia (NSIP); (iii) diffuse alveolar damage (DAD); (iv) organizing pneumonia (OP); (v) lymphoid interstitial pneumonia (LIP); and HP [6]. The diagnostic approach includes a multidisciplinary assessment based on clinical, radiological and histopathological data [6,12]. Some patients directly exhibit end-stage pulmonary fibrosis with unclassifiable ILD patterns [15].

Current ILD treatment includes anti-inflammatory and antifibrotic therapies, pulmonary rehabilitation, nutritional support, psychological care, and oxygen therapy. Lung transplantation may be considered for patients with progressive disease despite therapy [9,17]. Progressive pulmonary fibrosis (PPF) share pathophysiological features with IPF [2,9,18]. Data from the INBUILD study showed the efficacy and safety of nintedanib versus placebo in treating patients with fibrosing interstitial lung disease with a progressive phenotype [19,20].

Clinical management of ILD patients can be challenging. This pilot study aims to systematically examine the patient journey through the Brazilian and Mexican health systems. It assesses care models in each country, highlighting challenges, bottlenecks, unmet needs, and adherence to best practices in ILD reference centers in Brazil and Mexico.

## 2. Health System in Latin America—A Brief Description of the Health Systems in Brazil and Mexico

### 2.1. Brazilian Health System

Universal, equitable, and comprehensive access to health is ensured by the Brazilian Constitution [21]. It is essential to understand the complexities of fulfilling this constitutional premise in a country of continental dimensions with so many social and demographic inequalities [22]. In this context, the Brazilian health system is composed of state public management through the Unified Health System (SUS) associated with a private supplementary health system (Figure 1).

The public system is based on SUS, which is federally managed through the Ministry of Health. It provides a hybrid structure for health management with the simultaneous operation of a universal public service network and a complementary private sector [24]. The SUS is one of the world’s largest and most complex public health systems, including primary care and access to medium and high-complexity medical procedures such as blood donation or organ transplantation, thus ensuring full and free access for the entire Brazilian population. With SUS, the Ministry of Health aims to ensure full, equal, and universal access to the public health system, without discrimination, guaranteed by the Federal Constitution of 1988 [21]. SUS is financed with its own resources from the Union, States, and Municipalities, all of which are duly included in the Union’s budget. Brazilian taxpayers share the tax burden on medical care and the coverage of the federal universal health system for everyone, regardless of whether they are employed or unemployed [25].

The private health system is represented by private health insurance companies regulated by the National Agency for Supplementary Health (ANS). It is managed by companies that intermediary between the health system’s end users and service providers, using prepayment, health insurance, and compulsory contribution as financing [24].

### 2.2. Mexican Health System

The Mexican health system has seven different agencies divided into four sectors (Figure 2). It is a system segmented into several paying sources, public and private providers, organized under federal public institutions, private providers, and insurance companies. Thus, the organization of the Mexican health system varies according to financing, provision of health services and population coverage.

The social security system is composed of three national institutions that cover the formal salaried employees. The Mexican Institute of Social Security (Instituto Mexicano del Seguro Social—IMSS) covers employees in the private sector, and ISSSTE covers employees of the federal government.

The private sector, although regulated by the government, operates essentially independently. Private health insurance covers part of the population, mostly provided as an employment benefit provided by the private sector or by individual membership in higher income classes, in addition to the coverage by the IMSS or ISSSTE [23].

## 3. Methods

We conducted an evaluative study in reference centers for interstitial lung diseases in Brazil and Mexico in May 2021 with a panel of four medical center program coordinators with significant regional and population heterogeneities in the cities of Porto Alegre (Brazil), Feira de Santana (Brazil), Culiacán (Mexico) and Veracruz (Mexico). The medical center program coordinators, as well as professionals involved in the areas of access and advocacy. This model created a favorable environment for a better understanding of the processes related to the patient’s journey to encourage suggestions, share experiences, and solve problems.

### 3.1. Panel Study Methodology

This research design collected data from the same set of stakeholders at a single point in time. This approach allows researchers to observe insights into trends, relationships, and causality. To successfully compare the healthcare systems of Brazil and Mexico, we follow a robust methodology. Panel questions were previously formulated to investigate how healthcare access changes over time in both countries and analyze the impact of policy interventions on health outcomes. A representative non-probabilistic sample of healthcare facilities from Brazil and Mexico was selected to capture demographic variations, socioeconomic status, and geographical regions. A comparative qualitative analysis of patients’ journeys in the two countries aimed to identify patterns, trends, and differences in healthcare utilization, access, quality, and outcomes over time. Since data were collected at a single point of time, we have not employed appropriate statistical techniques to analyze the longitudinal data.

### 3.2. Data Acquisition

At first, four focal construction panels (workshops) were held with four pulmonologists specializing in ILDs. The choice of participants followed, as a homogeneity criterion, the institutional insertion of those involved and their role in the health system [26]. Each of the specialists was interviewed individually to understand better the journey of patients in each location and the determinants related to the functioning of each local health system, access to the specialized center, diagnosis, and treatment.

In a second moment, a workshop was conducted in a web conference format, where the same specialists had the opportunity to discuss and share their experiences in treating patients with ILD. This discussion addressed topics such as patient access and flow, the centers’ infrastructure, diagnosis, treatment, and multidisciplinary teams.

The meeting was held in an open discussion format, led by an external professional moderator. It was guided by standardized questions built after exploratory analysis extracted from the results of the first stage, in which panelists were encouraged to share experiences, problems, solutions, and controversies.

## 4. Results

### 4.1. Patient’s Journey

In both countries, the patient’s journey begins when they seek medical care after observing a chronic respiratory symptom.

In Brazil, the patient is admitted to the health system by a general practitioner from SUS or the supplementary health system. Current evidence indicates that inequalities in unmet healthcare needs reflect Brazil’s broader socioeconomic disparities [27]. Insufficient healthcare system funding creates geographic and logistic barriers to accessing specialized care. Patients with ILD often face significant delays between their initial consultation and the eventual diagnosis (Table 1). Those reporting unmet needs are predominantly among the poorest population segments and Brazil’s most economically disadvantaged regions. In the Brazilian health system, according to local specialists, up to 40% of patients have private health insurance, which provides easier access to complementary exams and specialized care; the other 60% depend exclusively on SUS, where there are usually barriers to quick access to more complex tests, such as HRCT and biopsy, as well as appointments with specialized physicians.

The Brazilian public centers for interstitial lung diseases included in this study have different management and funding systems, which can influence the treatment dynamics for these patients. Santa Casa de Misericórdia is a non-profit organization that receives resources from the Ministry of Health and state government, and the Hospital Estadual Cleriston Andrade is linked to the State University of Feira de Santana, which receives resources from SUS (Table 1).

In Mexico, two different realities were presented with variable autonomy and speed of resolution according to the funding source. However, access to specific treatment was similar in both ways. Access to specialized care is usually shorter in a navy hospital serving Mexican armed forces workers than in ISSSTE. When suspecting lung disease, the primary care physician requests a HRCT and forwards the patient to a pulmonologist in two weeks. In hospitals linked to the ISSSTE, the patient is initially evaluated by the primary care that, upon noticing an abnormality, requests a HRCT scan or chest X-ray. If radiological abnormalities are detected, the primary care physician consults with a specialist. Should further evaluation be required, the patient is referred to a pulmonologist.

In Mexico, the distance between hospitals and a referral center seems to influence access to a specialist. According to Mexican specialists, patients at institutions closer to the referral hospital may experience faster access to specialized care. The average waiting time for an evaluation by a pulmonologist is more than 30 days, and most patients have already undergone an HRCT scan.

Patients from other cities or even other Mexican states may experience a longer journey due to the bureaucratic access barriers to specialized centers. However, according to experts, telemedicine has been an extremely useful tool in this process, allowing physicians from different provinces to consult an ILD specialist. A face-to-face consultation can be promptly scheduled if the diagnostic suspicion is confirmed.

Both countries face a very similar reality regarding the long delay between initial symptoms and the diagnosis of ILD. Generally, these patients are admitted to the ILD referral centers at an advanced stage of the disease due to diagnostic complexity. This trend may be partially explained by ILD symptoms that are also common to other respiratory diseases such as asthma or COPD.

The patient journey begins at the primary care level (Figure 3), where a general practitioner usually sees the patient initially. In this scenario, interstitial pulmonary pathologies are often misdiagnosed as more prevalent illnesses such as COPD, asthma, or heart failure. This patient may receive inaccurate management until achieving a confident ILD diagnosis. Indeed, a crucial step in this patient’s trajectory is HRCT scanning. Thoracic radiologists can identify radiologic patterns of interstitial pulmonary involvement and suggest the diagnosis of ILD, which speeds up the patient’s referral to a specialist. According to experts, the elapsed time between the patient’s first contact with a doctor and the definitive diagnosis of ILD can be long. In Brazil, the diagnostic process can take from 6 to 18 months. In the Mexican system, the time for the patient to reach the specialist can vary from 1 to 12 months.

In ILD referral centers, the patient undergoes a clinical investigation aimed at underlying pathology that usually includes pulmonary function testing with lung volumes and diffusion capacity, autoantibody testing, a 6 min walk test (6MWT), and sometimes, a lung biopsy or a follow-up HRCT scan.

With the definitive diagnosis, the next step in this patient’s journey is access to adequate treatment, which may be challenging (Figure 3). At this specific point, there is a difference between Mexico and Brazil, as in the latter antifibrotic drugs are not yet available by SUS. Elevated cost is a major barrier to the direct purchase of antifibrotic drugs by most of the population. Most of the time, patients must resort to the judicial system to guarantee access to medication through SUS (Figure 3).

The judicialization consists of the search for treatments (medicines, exams, surgeries) through lawsuits. This occurs when the patient is unable to obtain these resources through the Unified Health System (SUS) or a private supplementary health system. The judicialization of health is characterized by the court’s claim of medicines developed to treat rare or highly complex clinical conditions that are not available in the public network or the supplementary health system.

In Brazil, the lack of availability of these drugs in SUS leads specialists to offer only supportive care, such as oxygen supplementation, until the patient can obtain the medication through litigation; differently, in Mexico, antifibrotic drugs have been available in different health sectors since 2018, and it may take from 1 week to 1 month until treatment is initiated.

### 4.2. Antifibrotic Therapy for ILD

The Brazilian National Health Surveillance Agency (ANVISA) has already approved both antifibrotics, Nintedanibe esilate and Pirfenidone [28,29], for use in Brazil. Still, the National Commission for the Incorporation of Technologies (CONITEC) [29] has not yet incorporated these antifibrotics into the public health system. According to experts, Brazilian patients have access to antifibrotics through litigation, a process that can take up to 8 months to complete.

In Mexico, antifibrotics were analyzed and incorporated [30] in different sectors of public health, thus making it possible to offer a broader antifibrotic treatment to patients with idiopathic pulmonary fibrosis (IPF).

In 2018, the Mexican government, through the Comisión Permanente del H. Congreso de la Unión, classified Idiopathic Pulmonary Fibrosis (IPF) as a catastrophic disease, supporting actions and strategies to prevent, detect and, when appropriate, provide timely treatment for this condition [31].

The classification of IPF as a catastrophic disease is due to its high cost of treatment associated not only with the direct costs of antifibrotic therapies but also with other associated expenses, such as nutritional and psychological therapy, pulmonary rehabilitation, supplementary oxygen, hospitalization, and the increased risk of acute exacerbations [30].

The Mexican coordinators explained that antifibrotic drugs can only be prescribed by a specialist inside the registered reference centers, thus allowing for greater control over prescriptions.

Mexican specialists highlighted that the treatment of IPF, the integration of the centers of reference in a network, and the consequent reduction in bureaucracy in prescribing antifibrotic drugs are key factors for the clinical care of IPF patients. Essential factors for proper patient care include staff familiarity with this disease and the necessary resources such as HRCT and pulmonary function tests. The Mexican specialists also emphasize that patients under antifibrotic treatment are exclusively monitored by pulmonologists improving patient safety.

About the decision-making regarding the use of the antifibrotic medication, we notice certain autonomy that may vary depending on the health sector. The Naval hospital receives a portion of the funds allocated to the armed forces by the federal government, and validation by a pulmonologist is required to request the use of antifibrotic for a patient. For most patients the decision takes place in a multidisciplinary discussion with a pulmonologist, a rheumatologist, a pathologist, a chest surgeon, and a radiologist. In hospitals linked to ISSSTE, which have a previously established budget, it is mandatory to rule out other alternative diagnoses, since IPF is an idiopathic disease, and to complete a checklist to allow the prescription of antifibrotics. All data regarding the diagnostic criteria must be validated by a central committee.

### 4.3. The Impact of the COVID-19 Pandemic on ILD Patients’ Journey

Brazil and Mexico experienced an unprecedented overload in their health systems during the COVID-19 pandemic [32]. Deficiencies in assistance to ILD patients were even more pronounced during the COVID-19 pandemic. In December 2019, a new strain of coronavirus emerged in China, initially named 2019-nCov and later renamed SARS-CoV2. The unprecedented speed of spread quickly transformed COVID-19 into a critical global public health problem with a devastating impact on healthcare services. Over the past few decades, Latin American health systems have chronically suffered from insufficient funding [33]. The precarious quality of care due to overloaded operations and structural deficiencies, even before the pandemic, suffered drastically during the exponential growth in demand for care for seriously ill patients during the pandemic. In this scenario, regional and socioeconomic inequalities were accentuated, as were barriers to accessing the public health system [34]. With restrictions on mobilization and social interaction, chronic outpatients with respiratory diseases, such as ILD, suffered a frank lack of assistance due to the emergency mobilization of health teams to support critically ill patients and the reasonable recommendation of home isolation for vulnerable populations.

We also observed an increase in the detection of new interstitial abnormalities during the COVID-19 pandemic [33]. There was an increased Imaging tests performed in patients with a suspected or confirmed diagnosis of COVID-19 have favored the identification of radiological findings compatible with ILD; as patients seek urgent medical care for suspected COVID-19, previous underlying interstitial disease could be detected early. The burden of post-COVID-19 fibrosis will become apparent with time, but early analysis from patients with COVID-19 on hospital discharge suggests that more than a third of recovered patients develop fibrotic abnormalities [35]. One study showed that 4% of patients with a disease duration of less than 1 week, 24% of patients with a disease duration of between weeks 1 and 3, and 61% of patients with a disease duration of greater than 3 weeks developed fibrosis [36].

In Brazil, COVID-19 profoundly impacted clinical follow-up and diagnostic procedures in ILD patients. With social isolation and lack of medical follow-up, the progression of the disease has been frequently described with a potentially higher risk of acute exacerbations, in addition to numerous cases of fibrosing disease secondary to COVID-19, including the need for specific outpatient clinics for the follow-up of these patients. There is particular concern that patients with COVID-19 remain symptomatic, a condition known as long-term COVID-19, and are at increased risk of bronchial remodeling and fibrosis [37]. The pandemic overloaded the health system and limited access to pulmonary medical specialists, thoracic imaging, and pulmonary function tests. This complex scenario favored further diagnostic delays, including new cases of interstitial disease that were not adequately evaluated.

In Mexico, patients were instructed to maintain social isolation at home with their medications dispensed in the hospital to a proxy. Enough antifibrotic medication has been dispensed for several months of treatment, and some patients have remote medical follow-ups by telemedicine.

## 5. Discussion

Brazil and Mexico share several similarities regarding the patient’s journey with interstitial disease. In both countries, the advanced stage of the disease is unfortunately common at the first evaluation at the referral center due to the diagnostic challenges, barriers to accessing specialized care, and high complexity testing. Additionally, the patient is initially evaluated by a general practitioner who is usually not trained to identify the signs of interstitial disease, favoring misdiagnoses with more prevalent diseases, such as asthma or COPD. Similar findings have been reported in other countries [38].

ILD is a debilitating condition that significantly impacts the quality of life and requires early recognition to enhance prognosis. Frequently, patients with ILD are misdiagnosed before being referred to a pulmonologist or ILD specialist clinic. In the INTENSITY survey, 55% of American adults with ILD reported receiving at least one misdiagnosis prior to their current diagnosis being confirmed [38]. This pre-diagnostic phase before reaching the referral center is crucial in the patient’s care pathway and represents a significant bottleneck in Latin America. Delayed recognition of ILD diagnosis in primary care is also a critical issue in the United States and Europe [39,40]. In Brazil, it typically takes twelve months for a patient to be referred to a pulmonologist; in Mexico, the average is eight months. Primary care providers should consider ILD in patients with chronic cough and dyspnea that remain unexplained after other common cardiovascular and respiratory conditions have been excluded. The delay in diagnosing and initiating treatment can lead to disease progression, declining quality of life, and significantly higher healthcare costs. Therefore, including a HRCT in the diagnostic work-up for patients suspected of having ILD could be beneficial.

The health systems of Brazil and Mexico exhibit notable differences in organization, funding, coverage, and service delivery. Each country’s unique socio-economic and political landscapes shape these distinctions, resulting in a distinct approach to healthcare provision (Table 2). One of the main differences lies in the funding mechanisms. Although hiring a private supplementary healthcare service in Brazil is possible, the Brazilian health system is based on the Unified Health System (SUS), which guarantees all citizens free access to healthcare services. Nevertheless, Mexico operates a segmented healthcare system where individuals may receive coverage through various channels: social security institutions, private insurance, or public healthcare services. This fragmented approach can lead to disparities in access based on employment status and income.

Coverage is another distinguishing factor. Brazilian SUS aims for universal coverage, with comprehensive services available to all citizens. However, due to resource limitations, service quality and availability disparities can still exist between regions. In contrast, Mexico’s system has multiple insurance schemes, including the Instituto Mexicano del Seguro Social (IMSS) and the Instituto de Seguridad y Servicios Sociales de los Trabajadores del Estado (ISSSTE), which provide coverage primarily to formal sector workers and government employees. This leaves a significant portion of the population, often in the informal sector, with limited access to healthcare services [25].

Service delivery approaches also diverge. Brazil emphasizes community-based care through the Family Health Program (Programa Saúde da Família), focusing on preventive and primary care. This approach helps to improve health outcomes, especially in remote and underserved areas [22]. While having a similar community healthcare model in place, Mexico also relies on a hospital-centric system, which can sometimes lead to overburdened facilities and challenges in providing accessible primary care services [25].

The decentralization of healthcare administration is another point of contrast. Brazil’s SUS is managed at the municipal, state, and federal levels, allowing for localized decision-making. While it has decentralized certain aspects, Mexico still maintains a more centralized governance structure. Cultural diversity and indigenous populations are some other characteristics that affect health policies [22,25]. In Mexico, indigenous communities often face cultural and linguistic barriers to accessing healthcare, while Brazil has established policies to address the health needs of its diverse population.

In summary, the Brazilian and Mexican health systems differ in terms of funding mechanisms, coverage models, and service delivery strategies. Brazil’s SUS prioritizes universal coverage and preventive care, while Mexico’s system encompasses a mix of social security, public, and private insurance schemes. Both systems face challenges in ensuring equitable access to quality healthcare, but their strategies reflect their respective societal and economic contexts.

Despite the similarities in the initial journey of patients with interstitial disease in both countries, the outcome tends to be different, largely due to access to treatment with the use of antifibrotics. While access to antifibrotics is not a major concern in the United States [40], reimbursement limitations significantly delay access to antifibrotic treatments in Europe. In some European countries, antifibrotic drugs are reimbursed only when patients receive a diagnosis from an ILD specialist center. Additionally, certain countries enforce criteria based on lung function and/or age before approving reimbursement [40]. The treatment of patients with interstitial disease in Brazil is hampered by restrictions on the direct dispensing of antifibrotics through SUS [27,28]. Despite being approved by ANVISA, their incorporation into SUS for the treatment of IPF was not approved by CONITEC to date [29]. The Brazilian Society of Pulmonology (SBPT) guidelines, published in 2012, reinforced the efficacy and safety of antifibrotic drugs in treating interstitial disease [2]. The long delay for treatment, especially in advanced disease with forced vital capacity (FVC) below 50%, can potentially impair the quality of life and increase the risk of exacerbations and progression of fibrosis [12,41,42].

The diagnosis and treatment of interstitial diseases also seem to have been affected by the COVID-19 pandemic that has raged worldwide since 2019. In addition to the delay in performing complementary exams and following up on patients, there were also new diagnoses of previous ILDs related to COVID-19.

These findings contribute to our understanding of the obstacles to early diagnosis and timely treatment in Brazil and Mexico. However, the study has some specific limitations. Firstly, the four ILD Tertiary Care Centers included in this research may not accurately represent conditions in other geographic areas. Therefore, caution should be exercised when generalizing these results to other ILD centers across Latin America. Secondly, individual patient data regarding the time elapsed between the onset of symptoms and the diagnosis of ILD, as well as the initiation of antifibrotic treatment, were unavailable, which may impact the study’s internal validity. Thirdly, data gathered from expert opinions in a panel study could increase the risk of information bias. Finally, the data were collected at a single point in time and might not capture longitudinal changes in a dynamic healthcare system. This research lays the groundwork for further studies in this area, including longitudinal investigations into the ILD patient journey across multiple centers in Latin America.

## 6. Conclusions

Brazil and Mexico have several similarities regarding the initial journey of the patient with interstitial disease; due to diagnosis difficulties, a patient may receive several different diagnoses until referred to a specialist. Despite the similarities in the initial journey of patients, the outcome tends to be different, mostly due to a difference in access to treatment with the use of antifibrotics.

Comprehending the patient’s healthcare journey is key to increasing access to proper treatment in middle-income countries. Socioeconomic factors, cultural nuances, and access barriers may influence effective treatments. These insights enhance healthcare delivery, improve outcomes, and promote equity in resource-constrained settings.

The integration and propagation of reference centers in ILD, as well as the reduction in bureaucracy in ordering high-cost medications, training of teams and greater access to diagnostic tests, can optimize the journey of patients with interstitial lung diseases. Furthermore, it is important to promote health policies for the adequate treatment of patients with interstitial disease, and for the sustainability and predictability of the health system.

## Figures and Tables

**Figure 1 ijerph-21-00647-f001:**
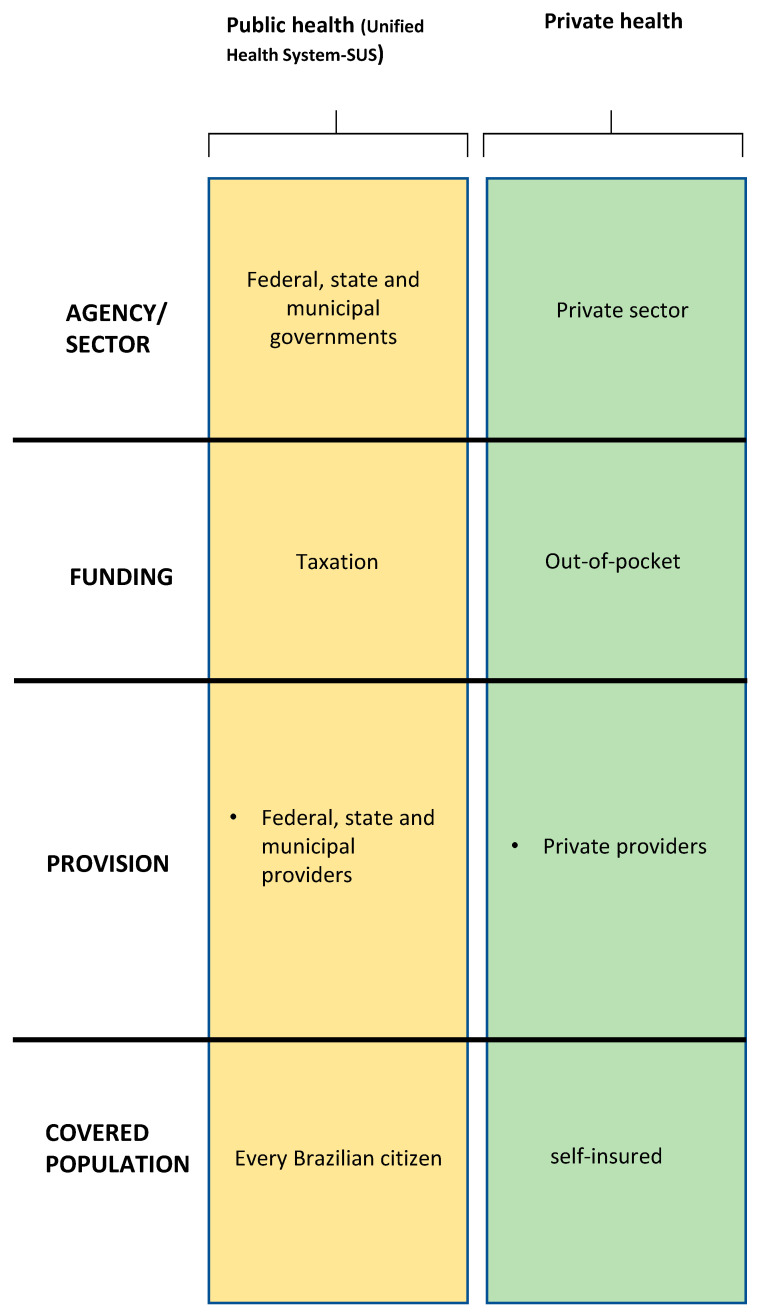
Organization of the Brazilian health system. Adapted from: Health Systems in Transition Vol. 22 No. 2 2020 [23].

**Figure 2 ijerph-21-00647-f002:**
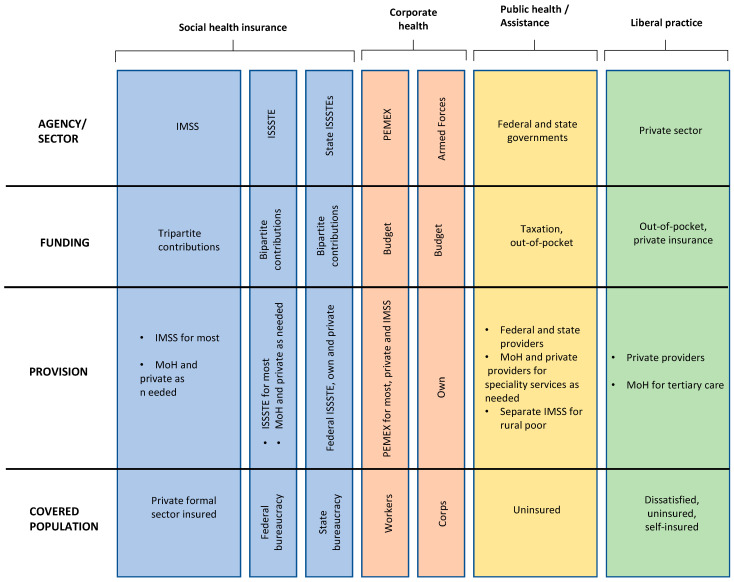
Organization of the Mexican health system. IMSS—Mexican Institute of Social Security, ISSSTE—Institute of Social Security and Services for State Employees, PMEX—Mexican Petroleum, MoH—Ministry of Health. Adapted from: Health Systems in Transition Vol. 22 No. 2 2020 [23].

**Figure 3 ijerph-21-00647-f003:**
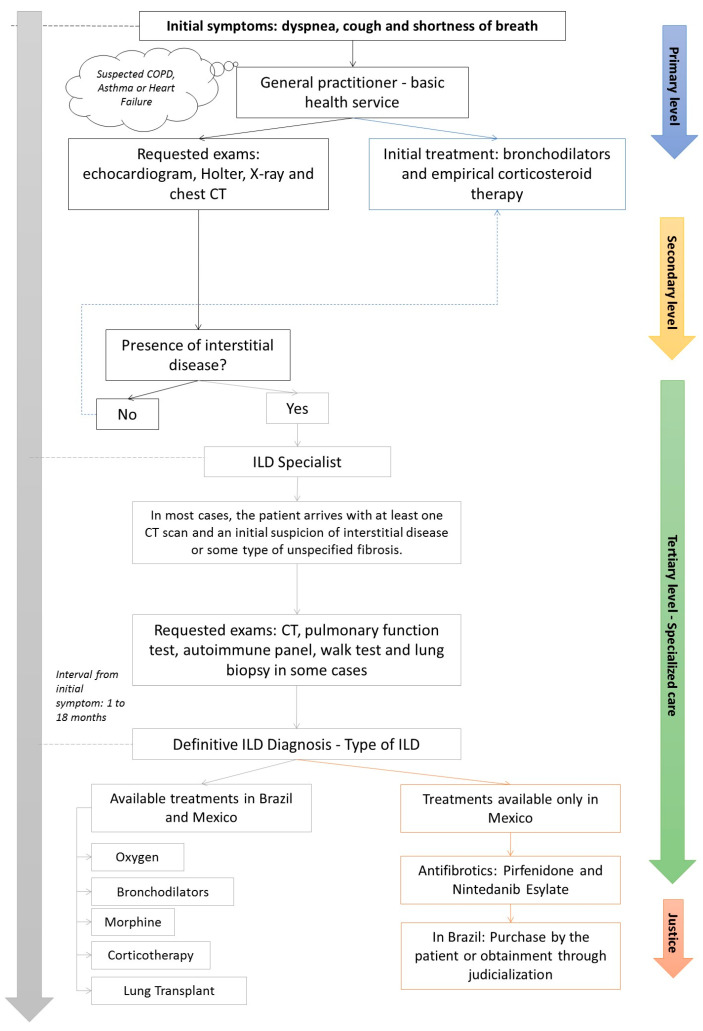
Figure 1 schematically shows the patient’s journey in treating ILD as reported by expert panel.

**Table 1 ijerph-21-00647-t001:** Characteristics of ILD Tertiary Care Centers.

	General Hospital	Philanthropic Hospital	ISSSTE Hospital	Navy Hospital
Geographic location	Feira de Santana, Northeast Brazil	Porto Alegre, South Brazil	Culiacán, Northwest Mexico	Veracruz, Southeast Mexico
ILD patients managed in a typical year	200	200	150	120
Elapsed time for referral in days	180–540	180–540	30–360	15–360
Guidelines or protocols followed by this institution	ATS/ERS/JRS/ALAT and SBPT guidelines	ATS/ERS/JRS/ALAT and SBPT guidelines	ATS/ERS/JRS/ALAT guideline	ATS/ERS/JRS/ALAT guideline
Multidisciplinary team for ILD patients at this institution	Pulmonologist, rheumatologist, thoracic radiologist, thoracic surgeon	Pulmonologist, rheumatologist, thoracic radiologist, lung pathologist, thoracic surgeon	Pulmonologist, rheumatologist, thoracic surgeon	Pulmonologist, rheumatologist, thoracic surgeon
Diagnostic workup	Spirometry, lung volumes, DLCO, 6MWT, chest X-ray, HRCT, serologic workup to detect autoimmune diseases, surgical lung biopsy, bronchoscopy	Spirometry, lung volumes, DLCO, 6MWT, HRCT, serologic workup to detect autoimmune diseases, surgical lung biopsy, bronchoscopy	Spirometry, lung volumes, DLCO, 6MWT, HRCT, serologic workup to detect autoimmune diseases, surgical lung biopsy, bronchoscopy	Spirometry, 6MWT, HRCT, serologic workup to detect autoimmune diseases, surgical lung biopsy, bronchoscopy
Treatments for ILD available on this institution	Corticosteroids, immunosuppressants, biologics, antifibrotic, oxygen, pulmonary rehabilitation	Corticosteroids, immunosuppressants, biologics, antifibrotic, oxygen, pulmonary rehabilitation, lung transplant	Corticosteroids, immunosuppressants, biologics, antifibrotic, oxygen, pulmonary rehabilitation	Corticosteroids, immunosuppressants, biologics, antifibrotic, oxygen, pulmonary rehabilitation
Average time in days (minimum, maximum) for commencing antifibrotics	120 (30, 240)	120 (60, 180)	15 (1, 60)	20 (7, 30)

ISSSTE: Instituto de Seguridad y Servicios Sociales de los Trabajadores del Estado; ILD: interstitial lung diseases; elapsed time for referral: duration between patient’s first signs or symptoms and the first appointment in specialty care in your institution; SBPT: Brazilian Respiratory Society; DLCO: diffusion capacity of lung for carbon monoxide; 6MWT: 6 min walk test (6MWT); HRCT: high resolution computerized tomography.

**Table 2 ijerph-21-00647-t002:** Differences between the Brazilian and Mexican Health systems.

	Brazilian Health System	Mexican Health System
Organization	Public healthcare services	Public healthcare services
	Private supplementary healthcare services	Social security institutions
		Private supplementary healthcare services
Coverage	Universal Coverage, with comprehensive services available to all citizens	IMSS and ISSSTE provide coverage primarily to formal sector workers and government employees, this leaves a significant portion of the population with a limited access to healthcare services
Service delivery	Brazil emphasizes community-based care through the Family Health Program (Programa Saude da Família), focusing on preventive and primary care. This approach helps to improve health outcomes, especially in remote and underserved areas	Mexico relies on a hospital-centric system, which could lead to overburdened facilities and challenges in providing accessible primary care services
Funding mechanisms	A decentralized administration. SUS is managed at the municipal, state, and federal levels, allowing for localized decision-making	Mexico maintains a more centralized governance structure

SUS—Sistema Unico de Saude; IMSS—Instituto Mexicano del Seguro Social; ISSSTE—Instituto de Seguridad yServicios Sociales de los Trabajadores del Estado.

## Data Availability

Additional data will be available upon request.

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
