# Peer review of "Improving Accessibility to Patients with Interstitial Lung Disease (ILD): Barriers to Early Diagnosis and Timely Treatment in Latin America"

_ijerph, 2024, doi:10.3390/ijerph21050647_

Round 1

Reviewer 1 Report

Comments and Suggestions for Authors

The study "Improving accessibility to patients with interstitial lung disease (ILD): Barriers to early diagnosis and timely treatment in Latin America" by Ricardo G. Figueiredo et al. examines the challenges faced by patients with interstitial lung disease (ILD) in Brazil and Mexico. It was found that, in both countries, patients often arrive at referral centers at an advanced stage of the disease due to the complexity of the diagnosis. However, differences in access to treatment are notable.

The authors have chosen a very relevant and important topic for their research. It is evident that they have invested a great deal of effort in collecting and analyzing the data. However, I have some observations on the submitted manuscript.

Introduction

Order and coherence in the presentation of information in the introduction are essential to provide a clear view of the context and objectives of the study. In this regard, the introduction lacks an organized and cohesive structure in addressing the relevant ILD issues. Although it begins with a general description of ILD, the transition to other aspects such as epidemiology, classification, diagnosis, and treatment is confusing and disorganized, making it difficult for the reader to understand. It is recommended that the authors revise and improve the wording of the introduction, following a logical sequence that first addresses a general description of ILD, followed by epidemiology, classification, diagnosis, and treatment, before presenting the specific objective of the study. This clearer and more organized structure will allow the reader to better understand the context and relevance of the work presented.

3. Methods

It would be beneficial for the readers' understanding to know the reference centers where the evaluative study was conducted. Clearly identifying these centers would provide additional context about the conduct of the study and the diversity of the populations or settings considered.

3.1. Panel study methodology

It would be important for the authors to report the sample size used in the evaluative study and its representativeness in relation to the epidemiology of the disease in the countries considered or worldwide. Knowing these details would allow assessment of the robustness of the findings and their applicability in different contexts.

4.1 Patients' Journey

In the results section, the authors mention that, in Brazil, patients are admitted to the health system by a general practitioner from the SUS or the complementary health system, which may result in a long interval between the first consultation and the final diagnosis of a PID. To support this assertion, it would be essential to provide details on how these data were confirmed. Is information available on the date of the first consultation of patients with chronic respiratory symptoms and the date they were properly diagnosed with ILD?

In the results section, the authors note that in a naval hospital that provides care to Mexican armed forces workers, access to specialty care tends to be shorter. However, it would be useful for readers to understand this in a comparative context: to what other health service is this shorter access time being compared? In addition, it would be relevant to know the average time in which this specialized care takes place at the aforementioned naval hospital.

Limitations

The authors state that the data were collected at a single point in time, and they have not employed appropriate statistical techniques to analyze longitudinal data.

Recognition of methodological limitations is crucial for accurate interpretation of the results. However, it would be beneficial for readers to understand how these limitations may have affected the interpretation of the findings and the generalizability of the results. Is the lack of longitudinal data considered to affect the internal validity or external applicability of the study? In addition, possible recommendations for future research to address this limitation, such as collecting longitudinal data for a more complete understanding of temporal trends, could be discussed. This additional analysis would help contextualize the contribution of the study and offer insights on how to advance research in this area.

Author Response

We are grateful to the reviewer for their insightful comments on our paper. We have incorporated changes to reflect most of the reviewer's suggestions. We attached a file with a point-by-point response to the reviewers' comments and concerns.

Reviewer 2 Report

Comments and Suggestions for Authors

I have thoroughly reviewed the manuscript titled "Improving accessibility to patients with z0 interstitial lung disease (ILD): Barriers to early diagnosis and timely treatment in Latin America." by Figueiredo et al. This study provides an important comparative analysis of the healthcare barriers that impact the diagnosis and treatment of ILD in Brazil and Mexico. However, it requires some revisions before the manuscript can be considered for publication.

1. Provide more details on the selection criteria and characteristics of the expert panelists, as well as the methods used for data collection and analysis.

2. Incorporate quantitative data or statistics on diagnostic delays, treatment access, and costs to strengthen the findings.

3. The section on COVID-19's impact on ILD diagnosis and treatment seems disconnected from the main focus of the study and could be removed or better integrated.

4. Limitations and potential biases of the study, such as the relatively small sample size or the reliance on expert opinions, should be addressed more explicitly.

Comments on the Quality of English Language

Minor editing required.

Author Response

(The authors gave the same response as above.)

Reviewer 3 Report

Comments and Suggestions for Authors

It would have been much more interesting to show the outcome of the diagnosed ILDs in Mexico and Brazil in comparison for example to US or Europe. The paper is more about the general weakness of the health systems - this is what we already believed to know.

Author Response

(The authors gave the same response as above.)

Round 2

Reviewer 1 Report

Comments and Suggestions for Authors

The authors have adequately addressed each of the comments and corrections made, which is reflected in the improvement of the manuscript. Therefore, this can be considered for publication.

Minor corrections:

In the discussion section, the authors take up the results obtained between patients from Brazil and Mexico and discuss the similarities in the trajectory of patients with interstitial disease. This information could be reinforced by comparing the data obtained in the study with that reported in other countries, in order to highlight the differences in the medical care of these patients. In addition, it could be emphasized how timely diagnosis and treatment can improve the patient's quality of life.

Author Response

Response to Reviewers

We are grateful to the reviewer for their insightful comments on our paper. We have incorporated changes to reflect most of the reviewer's suggestions. Here is a point-by-point response to the reviewers' comments and concerns.

The authors have adequately addressed each of the comments and corrections made, which is reflected in the improvement of the manuscript. Therefore, this can be considered for publication.

Minor corrections:

In the discussion section, the authors take up the results obtained between patients from Brazil and Mexico and discuss the similarities in the trajectory of patients with interstitial disease. This information could be reinforced by comparing the data obtained in the study with that reported in other countries, in order to highlight the differences in the medical care of these patients. In addition, it could be emphasized how timely diagnosis and treatment can improve the patient's quality of life.

Response: We agree with the reviewer that comparing our results with data reported in other countries would add relevance for the reader. We included these paragraphs in the discussion section and references to support this evidence as well, which we present below:

Page 12, line 9: “ILD is a debilitating condition that significantly impacts the quality of life and requires early recognition to enhance prognosis. Frequently, patients with ILD are misdiagnosed before being referred to a pulmonologist or ILD specialist clinic. In the INTENSITY survey, 55% of American adults with ILD reported receiving at least one misdiagnosis prior to their current diagnosis being confirmed40. This pre-diagnostic phase before reaching the referral center is crucial in the patient's care pathway and represents a significant bottleneck in Latin America. Delayed recognition of ILD diagnosis in primary care is also a critical issue in the United States and Europe41,42. In Brazil, it typically takes twelve months for a patient to be referred to a pulmonologist; in Mexico, the average is eight months. Primary care providers should consider ILD in patients with chronic cough and dyspnea that remain unexplained after other common cardiovascular and respiratory conditions have been excluded. The delay in diagnosing and initiating treatment can lead to disease progression, declining quality of life, and significantly higher healthcare costs43. Therefore, including a HRCT in the diagnostic work-up for patients suspected of having ILD could be beneficial.”

Page 13, line 9: “While access to antifibrotics is not a major concern in the United States42, reimbursement limitations significantly delay access to antifibrotic treatments in Europe. In some European countries, antifibrotic drugs are reimbursed only when patients receive a diagnosis from an ILD specialist center. Additionally, certain countries enforce criteria based on lung function and/or age before approving reimbursement42.

  1. Cosgrove GP, Bianchi P, Danese S, Lederer DJ. Barriers to timely diagnosis of interstitial lung disease in the real world: the INTENSITY survey. BMC Pulm Med. 2018 Jan 17;18(1):9. doi: 10.1186/s12890-017-0560-x. PMID: 29343236; PMCID: PMC5773175.
  2. Case AH, Beegle S, Hotchkin DL, Kaelin T, Kim HJ, Podolanczuk AJ, Ramaswamy M, Remolina C, Salvatore MM, Tu C, de Andrade JA. Defining the pathway to timely diagnosis and treatment of interstitial lung disease: a US Delphi survey. BMJ Open Respir Res. 2023 Nov 24;10(1):e001594. doi: 10.1136/bmjresp-2022-001594. PMID: 38007235; PMCID: PMC10680004.
  3. Moor CC, Wijsenbeek MS, Balestro E, Biondini D, Bondue B, Cottin V, Flewett R, Galvin L, Jones S, Molina-Molina M, Planas-Cerezales L, Prasse A, Prosch H, Russell AM, Viegas M, Wanke G, Wuyts W, Kreuter M, Bonella F. Gaps in care of patients living with pulmonary fibrosis: a joint patient and expert statement on the results of a Europe-wide survey. ERJ Open Res. 2019 Oct 21;5(4):00124-2019. doi: 10.1183/23120541.00124-2019. PMID: 31649949; PMCID: PMC6801215.
  4. Dempsey TM, Sangaralingham LR, Yao X, Sanghavi D, Shah ND, Limper AH. Clinical Effectiveness of Antifibrotic Medications for Idiopathic Pulmonary Fibrosis. Am J Respir Crit Care Med. 2019;200(2):168-174. doi:10.1164/rccm.201902-0456OC

Reviewer 2 Report

Comments and Suggestions for Authors

The authors have addressed all the concerns raised, demonstrating substantial improvements in the clarity, depth, and scientific rigor of the manuscript. Therefore, I recommend this manuscript for publication in the International Journal of Environmental Research and Public Health.

Comments on the Quality of English Language

Minor editing is required.

Author Response

We are grateful to the reviewer for their insightful comments on our paper.